# OpenReview forum: "RSafe: Incentivizing proactive reasoning to build robust and adaptive  LLM safeguards"
_NeurIPS.cc/2025/Conference — NeurIPS 2025 poster_

### Official Review · Reviewer_m19C · 2025-06-09

**Clarity:** 3
**Significance:** 2
**Originality:** 2
**Rating:** 2
**Confidence:** 4

**Summary:**

This paper presents RSafe, a two-stage, reasoning-based guard model designed to improve robustness and adaptability of LLM safety moderation. In Stage 1 (Guided Reasoning), RSafe elicits explicit, policy-aware step-by-step reasoning (enclosed in <think>…</think> tags) over a user-specified safety taxonomy; in Stage 2 (Reinforced Alignment), it applies rule-based reinforcement learning (GRPO) with format and accuracy rewards to refine reasoning paths for correct safety judgments.

**Questions:**

See weaknesses part.

**Ethical Concerns:**

["NO or VERY MINOR ethics concerns only"]

**Final Justification:**

At its core, the method applies basic GRPO on top of prompt engineering to enforce format compliance and classification correctness. This is an incremental advance, given that GRPO has known limitations: It uses 0/1 outcome reward with no supervision of intermediate reasoning steps, causing instability if a single sub‑step is erroneous.

Despite adopting reasoning-based training, the overall effectiveness seems to remain inferior to GuardReasoner‑8B.

**Quality:**

2

**Strengths And Weaknesses:**

Strengths
1. By grounding decisions in explicit, user-specified safety policies, RSafe generalizes beyond fixed training taxonomies to novel harmful categories and adversarial jailbreaks, a capability lacking in traditional guard models.


2. The enforced <think>…</think> schema yields verifiable reasoning traces at both train and test time.


Weaknesses
1. At its core, RSafe applies reinforcement learning with two straightforward reward components—format compliance and classification accuracy. While effective, this combination follows established training patterns with minimal adaptation.

2. The guided reasoning and RL are only conducted on Qwen2.5-7B. Weaker backbones may yield shallow rationales, limiting safety gains. A controlled comparison with GuardReasoner is needed.

3. The mechanism by which dynamic safety categories are integrated into training is described only at a high level. Critical implementation details are omitted. Providing a complete account of data curation and training schedules would improve reproducibility and allow readers to gauge practical overhead.

4. The adaptive mode presumes users can accurately define relevant safety taxonomies; in practice, overlooking critical policy categories (e.g., emergent threats) may degrade performance. No guidance or fallback is provided when policies are mis-specified.

---

> ### Author Rebuttal · Authors · 2025-07-31
>
> >### Comment 1: Straightfoward reward components with minimal adaptation
> *"At its core, RSafe applies reinforcement learning with two straightforward reward components—format compliance and classification accuracy. While effective, this combination follows established training patterns with minimal adaptation.*
>
> Thank you for raising this point. RSafe's 'minimal' reward design is not simplistic by oversight but by thoughtful choice: it provides a direct, noise-free approach to optimizing safety moderation objectives, avoiding reliance on intermediate reward models that could introduce noise or the pitfalls of complex reward designs. More specifically, Reinforcement learning (RL) algorithms are :
> 1. Highly sensitive to reward noise [1]. Adopting verifiable ground-truth labels avoids extra stochasticity that would otherwise be introduced by a learned reward model or external classifier.;
> 2. Prone to reward hacking, where the policy model can exploit flaws in the reward function rather than learning the intended behavior [2]. RSafe's concise verifiable reward surface leaves fewer unintended optima.
>
>  We outline several potential extensions below and explain why we approach them cautiously, ultimately returning to RSafe's straightforward design.
>
> | Candidate reward term     |Approach| Potential benefit   | Risks / drawbacks       |
> |-----------------------------------------------------------|-----------------------------------------------------------|----------------------------------------------------------------------------------|----------------------------------------------------------------------------------|
> |Difficluty-weighted reward | Stratify data according to difficulty levels, with additional bonuses for adversarial prompts.| Enhancing exploration on hard examples to improve robustness on edge cases. |Limited coverage may lead to overfitting to a narrow subset of adversarial examples.|
> | Policy-reference reward | Grant additional bonus if the reasoning path of rollout explicitly references the correct safety policy.| Stronger grounding in reasoning over the correct safety taxonomy. | Many harmful prompts violate multiple safety categories, thus introducing noise that can hinder moderation performance; policy model can exploit category frequencies rather than reasoning over safety context. |
> |Calibration reward | Penalize models when they output incorrect answers with high confidence or correct answers with low confidence. |  Incentivizing calibrated predictions, mitigating guard model's hallucinations under out-of-distribution safety domains and thereby improving robustness.| Approaches for obtaining confidence estimates can introduce additional inaccuracies thus may lead to reward hacking [3].|
>
>
>
> >### Comment 2: Reliance on the capabilities of the backbone model
> *"The guided reasoning and RL are only conducted on Qwen2.5-7B."*
>
> Thank you for raising this point. We have added experiments using Qwen2.5-3B and Llama3.1-8B as alternative backbones, with updated results for effectiveness and robustness shown in Table 1 and Table 2. The results confirm that the choice of backbone influences performance: Qwen2.5-3B performs worse than Qwen2.5-7B, while Llama3.1-8B performs slightly worse on effectiveness but achieves higher robustness. These results suggest that RSafe can better leverage stronger backbone models to enhance safety through more robust reasoning capabilities.
>
> Table 1. Performance comparison on safety moderation benchmarks across different backbone models.
>
> ||         | ToxicChat  |            | OpenAI Mod |            | Aegis       |            | SafeRLHF   |            | BeaverTails |            | XSTEST      |            | Weighted Avg |               |
> |------------------------|------------------------|------------|------------|------------|------------|-------------|------------|------------|------------|-------------|------------|-------------|------------|--------------|---------------|
> | **Method** |        **Backbone model**              | Acc        | F1         | Acc        | F1         | Acc         | F1         | Acc        | F1         | Acc         | F1         | Acc         | F1         | Acc          | F1            |
> | RSafe| Qwen2.5-7B-Instruct| 0.957      | 0.728      | 0.817      | 0.764      | 0.834       | 0.849      | 0.914      | 0.921      | 0.814       | 0.832      | 0.970       | 0.816      | 0.904        | 0.889         |
> |RSafe| Qwen2.5-3B-Instruct| 0.947      | 0.692      | 0.795      | 0.739      | 0.789       | 0.800      | 0.842      | 0.854      | 0.722       | 0.745      | 0.806       | 0.407      | 0.839        | 0.820         |
> |RSafe| Llama3.1-8B        | 0.934      | 0.648      | 0.739      | 0.698      | 0.812       | 0.833      | 0.910      | 0.918      | 0.824       | 0.841      | 0.914       | 0.617      | 0.890        | 0.878         |
>
> Table 2. Robustness comparison on WildGuardTest across different backbone models.
>
> |          |         | Overall     |           | Adversarial |           | Vanilla     |           |
> |------------------------|----------------|-------------|-----------|-------------|-----------|-------------|-----------|
> |     **Backbone Model**   |    **Method**           | Acc         | F1        | Acc         | F1        | Acc         | F1        |
> | Qwen2.5-7B-Instruct    | RSafe          | 0.828       | 0.772     | 0.779       | 0.668     | 0.871       | 0.845     |
> |                        | RSafe-adaptive | 0.845       | 0.823     | 0.791       | 0.717     | 0.911       | 0.899     |
> | Qwen2.5-3B-Instruct    | RSafe          | 0.807       | 0.754     | 0.772       | 0.693     | 0.839       | 0.802     |
> |                        | RSafe-adaptive | 0.830       | 0.800     | 0.769       | 0.717     | 0.883       | 0.868     |
> | Llama3.1-8B            | RSafe          | 0.849       | 0.815     | 0.789       | 0.715     | 0.901       | 0.888     |
> |                        | RSafe-adaptive | 0.861       | 0.838     | 0.802       | 0.747     | 0.912       | 0.905     |
>
>
>
> >### Comment 3: Disclosure of training details
> *"The mechanism by which dynamic safety categories are integrated into training is described only at a high level."*
>
> Thank you for raising this issue. During training, we used a default instruction prompt set providing in-distribution data to teach the model context-aware reasoning based on given safety policies. At inference, we support dynamic safety categories, allowing users to specify their required safety policies based on their needs.
>
> To improve reproducibility and address practical overhead, we provide detailed training data scale and compute usage in Appendix C.3. In total, we curated a training dataset of approximately 12,000 examples, balanced across harmful and unharmful samples from five safety-related corpora, covering both prompt-level and response-level harmfulness detection tasks across diverse safety scenarios.
>
> For computational resources, we used the VERL codebase on 4×A100 80GB GPUs with a batch size of 128 and a maximum sequence length of 2048 tokens. Each run was trained for 3 epochs with rollout as 4, taking approximately 3–4 hours.
>
> >### Comment 4: Missing / mis-specified safety taxonomies
> *"The adaptive mode presumes users can accurately define relevant safety taxonomies; in practice, overlooking critical policy categories (e.g., emergent threats) may degrade performance."*
>
> Thanks for bringing up this problem. When users encounter novel safety situations and cannot supply a clear policy, RSafe provides an option to omit the safety taxonomy while still providing strong moderation. Table 4 presents this configuration as “RSafe (w/o category specification).”, where it outperforms its baselines and the default RSafe version. RSafe can draws on its internal safety self-awareness [5] and the context-aware reasoning capability acquired during training to provide flexible and accurate decisions, instead of trying to fit new hazards into specified safety categories that may be ill-suited.
> These results demonstrate that RSafe remains both robust and protective even when users cannot specify a taxonomy and the scenario lies outside all predefined categories.
>
>
> Table 4. RSafe's performance without safety category specification compared to its adaptive mode and other baselines on out-of-domain benchmarks featuring novel safety categories.
>
> |              | WildGuardTest |              | MedSafetyBench | CARES (base) |              | CARES (adv)  |              |
> |-------------------------|--------------|--------------|----------------|--------------|--------------|--------------|--------------|
> |                         | Acc          | F1           | Acc            | Acc          | F1           | Acc          | F1           |
> |  OpenAI Moderation|0.583 | 0.122|0.006|0.241|0.069|0.220 |0.018
> | ShieldGemma-9B|0.731  |0.590|0.288|0.475|0.504|0.324|0.247|
> |  LlamaGuard3-8B|0.822  |0.763 |0.644|0.646|0.711|0.382|0.354|
> | RSafe               |    0.819     |  0.758           | 0.588          | 0.576        | 0.631        | 0.433        | 0.429        |
> | RSafe-adaptive      |    0.840         |     0.797        | 0.804          | 0.709        | 0.773        | 0.501        | 0.535        |
> | **RSafe (w/o category specification)**    |   0.835          |  0.792           |  0.691        |   0.643      | 0.707       |  0.454      |0.475       |
>
>
> References
>
> [1] Liang, Xize, et al. "ROPO: Robust Preference Optimization for Large Language Models."  (ICML 2024).
>
> [2] Fu, Jiayi, et al. "Reward shaping to mitigate reward hacking in rlhf."  (2025).
>
> [3] Mroueh, Youssef. "Reinforcement Learning with Verifiable Rewards: GRPO's Effective Loss, Dynamics, and Success Amplification." (2025).
>
> [4] Lin, Yuping, et al. "Towards understanding jailbreak attacks in llms: A representation space analysis." (2024).

---

> > ### Comment · Reviewer_m19C · 2025-08-03
> >
> > At its core, the method applies **basic GRPO** on top of prompt engineering to enforce format compliance and classification correctness. This is an incremental advance, given that GRPO has known limitations:
> > It uses 0/1 outcome reward with no supervision of intermediate reasoning steps, causing instability if a single sub‑step is erroneous.
> >
> > Despite adopting reasoning-based training, the overall effectiveness seems to remain inferior to GuardReasoner‑8B.
> >
> > My comment about the model failing when users cannot specify harmful categories was misunderstood, my concern was that the model’s reliance on user‑specified categories could undermine its moderation capability in practice.

---

> ### Author Response · Authors · 2025-08-05
>
> Thanks for your time and your comment, regarding your concerns:
> 1. The concern that the lack of supervision over intermediate steps might introduce instability does not hold in our setting. Through manual inspection, we repeatedly confirm the consistency between the reasoning path and the final prediction. Moreover, eliminating intermediate supervision mitigates the risks of label noise and reduces annotation burden, ultimately enhancing the robustness of our framework.
>
> 2. RSafe, trained on Qwen2.5-7B with approximately 10k examples, achieves competitive overall effectiveness compared to GuardReasoner, which is trained on Llama3.1-8B with 127k examples. For a controlled comparison, we conducted experiments using the same backbone model and training datasets, with the only difference being the training pipeline. The results, presented in Table 1 and Table 2, demonstrate that RSafe's superior performance.
>
> Table 1. Performance comparison on safety moderation benchmarks across different methods.
>
> | **Backbone**          | **Method**       | ToxicChat Acc | F1   | OpenAI Mod Acc | F1   | Aegis Acc | F1   | SafeRLHF Acc | F1   | BeaverTails Acc | F1   | XSTEST Acc | F1   | Weighted Avg Acc | F1    |
> |-----------------------|------------------|---------------|------|----------------|------|-----------|------|----------------|------|------------------|------|--------------|------|------------------|-------|
> | Qwen2.5-7B-Instruct   | RSafe            | 0.957         | 0.728| 0.817          | 0.764| 0.834     | 0.849| 0.914          | 0.921| 0.814            | 0.832| 0.970        | 0.816| 0.904              | **0.889** |
> | Qwen2.5-7B-Instruct   | GuardReasoner    | 0.937         | 0.665| 0.792          | 0.730| 0.819     | 0.837| 0.901          | 0.911| 0.820            | 0.842| 0.899        | 0.579| 0.887              | **0.827** |
>
> Table 2. Robustness comparison on WildGuardTest across different methods.
>
> | **Backbone**          | **Method**       | Overall Acc | F1     | Adversarial Acc | F1   | Vanilla Acc | F1   |
> |-----------------------|------------------|-------------|--------|------------------|------|--------------|------|
> | Qwen2.5-7B-Instruct   | RSafe            | 0.828       | 0.772  | 0.779            | 0.668| 0.871        | 0.845|
> | Qwen2.5-7B-Instruct   | RSafe-adaptive   | 0.845       | **0.823**| 0.791          | 0.717| 0.911        | 0.899|
> | Qwen2.5-7B-Instruct   | GuardReasoner    | 0.822       | **0.782**| 0.772          | 0.673| 0.875        | 0.859|
>
>
>
> 3. While providing accurate and fine-grained safety categories yields the best results, RSafe remains robust even without such specifications, according to results shown in Table 4 in the rebuttal response. Deliberately specifying incorrect or contradictory categories falls outside the scope of our intended use and is thus not addressed in this work.

---

### Official Review · Reviewer_A8Ug · 2025-07-03

**Clarity:** 3
**Significance:** 2
**Originality:** 3
**Rating:** 4
**Confidence:** 3

**Summary:**

The paper introduces RSafe, a system‑level guard model for LLMs that (i) adds an explicit reason‑then‑predict stage (“Guided Reasoning”) and (ii) aligns this behaviour with a lightweight rule‑based reinforcement‑learning procedure (“Reinforced Alignment”, based on GRPO). This exhibits generalisation to emerging harmful categories and jailbreak attacks (WildGuardTest). At inference time, users can supply arbitrary safety policies; RSafe reasons over the supplied categories without retraining. The rationale is also always returned, enabling interpretability in the guard model.

**Questions:**

The primary claim of the paper is its ability to accept various user-supplied safety taxonomies, **allowing developers to include and exclude safety policies at will**. This is mentioned as one of the most significant contributions of the paper on page 3. However, the experimental results do not necessarily investigate this claim in depth, beyond finding improved performance on a pre-established adversarial examples dataset.

Many questions I would like to see answered are around providing experimental evidence to strengthen or weaken this claim. Primarily:
1. One evaluation that would be interesting would be to see how RSafe handle entirely idiosyncratic, edge-case, or otherwise highly specific user‑defined safety rules, particularly when compared with GPT‑4o or Claude.
2. How does RSafe work when certain safety metrics are excluded (e.g. are users/developers able to exclude copyright if they do not care about this specific risk, and "remove refusals" related to copyright?). Can it delineate and clearly change safety taxonomies compared with other baseline guard options?
3. Is there possibility to expand the testing of robustness beyond WildGuardTest (Section 4.2), toward evals that demonstrate that it can follow instructions with respect to novel safety criteria?

Both positive and negative results would strengthen my confidence and rating.

**Ethical Concerns:**

["NO or VERY MINOR ethics concerns only"]

**Final Justification:**

The issue I raised in my review -- allowing developers to include and exclude safety policies at will -- was resolved. I am not providing a stronger rating because RSafe's performance in this area is good but not stellar. In the results shown in the rebuttal, RSafe is highly comparable with 4o's performance.

**Limitations:**

The main limitation has already been mentioned above -- centering around addressing the claim that the safety policy of RSafe can be dynamically aligned with task-specific requirements.

**Quality:**

3

**Strengths And Weaknesses:**

**Strengths:**
The primary strength seems to be that RSafe addresses, through a unique RL objective, is the need for guardrail models that adapt with different safety policies and user-specified safety taxonomies by accepting arbitrary user‑supplied safety taxonomies at runtime. This drop‑in policy swap is done without retraining; a capability missing in prior guard models. This can generalize to various safety violation scenarios, such as intellectual property, which are more out-of-distribution. Furthermore, the reasoning is done for each safety policy.

**Weaknesses:**
- The experiments do not test the adaptiveness of the policy, despite this being the primary advantage that is proclaimed by RSafe -- see "Questions" and "Limitations" for more experiments I'd like to see done. While there was an evaluation on WildGuardTest, which shows increased performance on out-of-distribution adversarial examples, this is distinct from being able to show that safety policies can change and RSafe can adapt to different safety needs.
- The experiments shown do not demonstrate a significant performance improvement over other open-source guardrail models.

---

> ### Author Rebuttal · Authors · 2025-07-31
>
> >### Comment 1: Robustness on edge cases / adaptiveness to highly specific safety rules
> *"One evaluation that would be interesting would be to see how RSafe handle entirely idiosyncratic, edge-case, or otherwise highly specific user‑defined safety rules, particularly when compared with GPT‑4o or Claude.""*
>
> Thank you for raising this interesting question. We interpret your suggestion as encompassing two aspects: (1) robustness evaluation under edge-case adversarial prompts, such as those from red-teaming scenarios, and (2) adaptivity evaluation under highly specific user-defined safety rules. We conducted two separate experiments accordingly.
>
> For the first aspect, we focused on edge-case prompts from two red-teaming datasets: JailbreakBench [1] and JailJudge [2]. These datasets include jailbreak attacks (e.g., PAIR [3], GPTFuzzer [4]) and deceptive strategies such as role-playing and hypothetical scenarios, as well as prompts based on real-world risks.
> To ensure the selected prompts represent genuinely challenging safety risks that are not trivially detectable, we filtered out prompts that GPT-4o refused outright with generic rejections like *"I'm sorry, I can't help with..."*. After filtering, 17 prompts from JailbreakBench and 432 from JailJudge remained.
>
> We then used RSafe and GPT-4o with identical instruction prompts to perform safety moderation. As shown in Table 1, RSafe performed slightly better than GPT-4o on JailJudge but worse on JailbreakBench, likely due to the small sample size in the latter making the difference appear more pronounced. While RSafe's accuracy in detecting safety risks drops compared to its ~0.9 average on in-domain general safety benchmarks (calculated from Table 1 in the original paper), it maintains strong protective capability and matches GPT-4o when used as *LLM-as-judge*, demonstrating robustness among edge cases.
>
> Table 1. Robustness on edge cases: RSafe vs. GPT-4o.
>
> |      | Jailbreakbench (\#17)   | Jailjudge (\#432) |
> |-------------|-------------|-------------:|
> |         |     Acc   |    Acc     |
> | GPT‑4o        |  0.882      | 0.682   |
> | RSafe       |    0.706  |   0.694   |
>
> For the second experiment evaluating adaptiveness, we follow the setup from Aegis 2.0 [5] to synthetically generate prompts that either violate or comply with fine-grained guidelines across three domains: Financial Advice, Legal Advice, and Medical Advice. We construct dataset with 40 prompts per category, balanced between violations and compliant cases.
>
> As shown in Table 2, both RSafe-adaptive and GPT-4o achieve near-perfect accuracy. This result is expected: the policies are clearly defined, and the synthetic prompts are tightly aligned with the policy framing, making this task primarily a test of instruction-following rather than nuanced reasoning. By contrast, the adversarial prompts in Table 1, drawn from red-teaming attacks, are significantly more challenging and more likely to bypass safety defenses. This highlights the importance of focusing on adversarial/edge-case examples when evaluating robustness under highly specific, user-defined safety scenarios.
>
> Table 2. Adaptiveness on highly user-specific rules: RSafe vs. GPT-4o.
>
> |            | Financial Advice （\#40） |            | Law Advice （\#40） |            | Medical Advice （\#40）|            |
> |------------|------------------|------------|------------|------------|----------------|------------|
> |        | Acc | F1 | Acc | F1 | Acc | F1 |
> | GPT-4o |   1  | 1   |   1  | 1   |    1 |  1  |
> | RSafe-adaptive |  1 | 1   |  0.975   | 0.974   |  0.975   |  0.974  |
>
>
> >### Comment 2: Adaptiveness under safety category exclusion
> *"How does RSafe work when certain safety metrics are excluded""*
>
> Thank you for bringing this up. RSafe-adaptive allows users to specify safety policies during inference, enabling the exclusion of certain policies when specific risks are not relevant. Based on your suggestion, we added evaluations to test RSafe's ability to exclude safety policies.
>
> We used three benchmarks with annotated safety taxonomies: Sorry-Bench, HarmBench, and WildGuardTest. We evaluated two settings: (1) RSafe-adaptive with full taxonomy adherence, and (2) RSafe-adaptive (w/ removal) that exclude specified categories. For example, in Sorry-Bench, we removed five subcategories under *Hate Speech Generation*, leaving 40 subcategories, then examined performance on both removed and remaining categories. GPT-4o was included for comparison using identical prompts.
>
> As shown in Table 3, excluding categories had no significant impact on remaining category performance. However, both RSafe and GPT-4o showed limited ability to remove refusals in excluded categories.
>
> Investigation revealed that prompts failing to be excluded often violated multiple taxonomies simultaneously. For instance, prompts annotated as hate speech frequently triggered other safety categories due to policy overlaps. Even after removing *Hate Speech Generation*, RSafe continued flagging these prompts under different taxonomies.
> Manual inspection across all benchmarks confirmed this overlap was consistently the cause. Successfully excluded prompts violated only the removed category.
>
> The result demonstrates RSafe's policy exclusion capability. However, safety taxonomies are not disjoint, and prompts often violate multiple categories, making complete exclusion non-trivial.
>
> Table 3. Adaptiveness comparison on exclusion tasks for RSafe and GPT-4o.
>
> |                        | SORRY-Bench                        |            | HarmBench                             |            |            | WildGuardTest           |            |
> |--------------------------------|------------------------------------|------------|----------------------------------------|------------|------------|--------------------------|------------|
> |                                | Hate Speech Gen (#50) | Others (#400) | Misinformation (#65) | Others (#335) | Copyright (#30) | Others (#1659) |          |
> |                                | Acc                  | Acc         | Acc                 | Acc          | Acc           | Acc              | F1         |
> | GPT-4o - adaptive       | 0.980                | 0.780       | 0.862               | 0.976        | 0.700         | 0.849            | 0.825      |
> | GPT-4o-adaptive (w/ removal) | 0.940                | 0.783       | 0.707               | 0.964        | 0.633         | 0.856            | 0.834      |
> | RSafe - adaptive               | 1.000                | 0.728       | 0.769               | 0.749        | 0.400         | 0.860            | 0.822      |
> | RSafe-adaptive (w/ removal)         | 1.000                | 0.730       | 0.646               | 0.728        | 0.367         | 0.847            | 0.802      |
>
>
> >### Comment 3: Expand robustness evaluation
> *"Is there possibility to expand the testing of robustness beyond WildGuardTest (Section 4.2), toward evals that demonstrate that it can follow instructions with respect to novel safety criteria?"*
>
> Thank you for raising this point. We expanded the original Section 4.2 experiments on WildGuardTest to validate RSafe's adaptivity to novel safety criteria by adding two medical ethics benchmarks: MedSafetyBench and CARES, with CARES including both base and adversarial versions. As shown in Table 3, OpenAI Moderation, ShieldGemma-9B, LlamaGuard3-8B, and RSafe all exhibit significant accuracy drops on these datasets compared to their average performance on general safety benchmarks (results from Table 1 in the original paper). This indicates that MedSafetyBench and CARES present out-of-distribution safety challenges beyond all models' training domains.
>
> As shown in Table 3 below, RSafe outperforms baseline models on both datasets. When guided by specific medical principles at inference (RSafe-adaptive), performance improves notably. Compared to LLM-as-judge (GPT-4o with corresponding ethical guidelines), RSafe-adaptive achieves comparable performance, demonstrating strong adaptiveness and reliability as a safety guardrail model.
>
> Table 3. Adaptiveness comparison of RSafe and baseline models on novel safety topics.
>
> | Guard Model             | Weighted Avg (general) | MedSafetyBench | CARES (base) |              | CARES (adv)  |              |
> |-------------------------|------------------------|----------------|--------------|--------------|--------------|--------------|
> |                         | Acc                    | Acc            | Acc          | F1           | Acc          | F1           |
> | OpenAI Moderation       | 0.626                  | 0.006          | 0.241        | 0.069        | 0.220        | 0.018        |
> | ShieldGemma-9B          | 0.702                  | 0.288          | 0.475        | 0.500        | 0.324        | 0.247        |
> | LlamaGuard3-8B          | 0.877                  | 0.644          | 0.646        | 0.711        | 0.382        | 0.354        |
> | **RSafe**               | 0.904                  | 0.588          | 0.576        | 0.631        | 0.433        | 0.429        |
> | **RSafe-adaptive**      | -                      | 0.804          | 0.709        | 0.773        | 0.501        | 0.535        |
> | LLM-as-judge (GPT-4o)   | -                      | 0.881          | 0.632        | 0.738        | 0.456        | 0.527        |
>
> References
>
> [1] Chao, Patrick, et al. "Jailbreakbench: An open robustness benchmark for jailbreaking large language models." (NIPS 2024).
>
> [2] Liu, Fan, et al. "Jailjudge: A comprehensive jailbreak judge benchmark with multi-agent enhanced explanation evaluation framework." (2024).
>
> [3] Chao, Patrick, et al. "Jailbreaking black box large language models in twenty queries." (2023).
>
> [4] Yu, Jiahao, et al. "Gptfuzzer: Red teaming large language models with auto-generated jailbreak prompts." (2023).
>
> [5] Ghosh, Shaona, et al. "Aegis2.0: A diverse ai safety dataset and risks taxonomy for alignment of llm guardrails." (2025).

---

> > ### Comment · Reviewer_A8Ug · 2025-08-07
> >
> > Thank you for the detailed rebuttal. I will raise my score.

---

> > > ### Author Response · Authors · 2025-08-07
> > >
> > > Thanks for your time and your acknowledgement.

---

### Official Review · Reviewer_Fe4b · 2025-07-06

**Clarity:** 4
**Significance:** 4
**Originality:** 2
**Rating:** 4
**Confidence:** 5

**Summary:**

The paper proposes RSafe – an adaptive safeguard for large language models that first performs policy-guided, safety reasoning and then applies rule-based reinforcement learning to align those reasoning paths with correct safety judgments. This two-stage paradigm enables the model to generalize across different safety principles rather than merely memorize labels, letting it moderate both prompts and responses under user-specified policies. Experiment results demonstrate its effectiveness across different benchmarks compared to baselines such as Llama Guard and ShieldGemma.

**Questions:**

Please respond to the weaknesses above.

**Ethical Concerns:**

["NO or VERY MINOR ethics concerns only"]

**Final Justification:**

My major concerns are resolved. However, other reviewers also have questions regarding the adaptiveness of the proposed method and reviewer A8Ug's point on reasoning process reward supervision remain valid. Thus, I will keep my score.

**Quality:**

3

**Strengths And Weaknesses:**

**Strengths**

1. The paper introduces an adaptive safeguard that couples policy-guided, step-by-step safety reasoning with rule-based policy optimization, producing a method that is both promising and empirically effective.

2. The manuscript is clear and well structured: the problem statement, algorithmic details, and experimental protocols are presented in a logical sequence, making the work easy to follow.

3. The evaluation is comprehensive. It spans several model scales, safety benchmarks, and jailbreak settings, and the authors provide both quantitative metrics and qualitative analyses that support their claims.

**Weaknesses**

1. The proposed formulation is closely related to the GuardReasoner framework. The principal difference is that GuardReasoner relies on DPO, whereas the present work adopts GRPO with verifiable rewards such as accuracy and format compliance. This change, while sensible, may appear incremental and therefore limits the novelty of the contribution.

2. In the robustness experiment on unseen safety categories, the “Criminal Planning / Confessions” topic can overlap with categories already covered by Llama-Guard (violent crime) and ShieldGemma (dangerous content). To better demonstrate adaptiveness to truly novel topics, I suggest (i) choosing a domain unrelated to safety—e.g., a keyword-mentioning or mathematics discussion task—and (ii) adding an LLM-as-judge baseline (such as GPT-4o) to evaluate safeguard reliability when judgment is delegated to an external model.

---

> ### Author Rebuttal · Authors · 2025-07-30
>
> >### Comment 1: Comparison between DPO and GRPO
> *"The proposed formulation is closely related to the GuardReasoner framework. The principal difference is that GuardReasoner relies on DPO, whereas the present work adopts GRPO with verifiable rewards such as accuracy and format compliance. This change, while sensible, may appear incremental and therefore limits the novelty of the contribution."*
>
> Thank you for your thoughtful comment. While the transition from DPO to GRPO may initially seem like a natural extension, from pairwise preference learning to verifiable reward optimization, and merely a change in the loss function, GRPO in fact introduces several essential capabilities that go beyond this surface-level shift. Specifically, GRPO enables more effective alignment in safety moderation tasks by leveraging (1) verifiable supervision, (2) multi-objective reward integration, and (3) on-policy optimization. These properties make GRPO particularly well-suited for safety moderations, as explained in details below.
>
> | Aspect                     | DPO                      | GRPO                        | Advantage                                   |
> |---------------------------|-----------------------------------------------------------|----------------------------------------------------------------------------------|----------------------------------------------------------------------------------|
> | **Supervision source**     | Requires reasoning process data specific to safety moderation tasks   | Requires only the ground-truth safety label for the prompt or response     | Reduces reliance on reasoning process data,  matches well with the rule-checkable safety moderation task                              |
> | **Multi-objective reward** | Not natively supported                                   | Supports multiple reward dimensions                                   | Currently, the rewards we incorporate are format and accuracy. As specific needs require, we can add rewards in dimensions such as fluency, helpfulness, length and confidence to guide the generation of reasoning process.           |
> | **Robustness to OOD data**   | Offline, off-policy : relies on offline preference pairs, susceptible to out-of-distribution (OOD) data [4][5] | Online, on-policy :  generates training data dynamically through rollouts from the current policy, allowing continuous adaptation and stronger generalization to out-of-distribution safety scenarios [6]. |  GRPO’s on-policy optimization that support dynamic exploration beyond static supervision aligns naturally with the demands of safety moderation, which requires robust generalization to unseen jailbreaks and evolving harms. |
>
>
>
> >### Comment 2: Demonstrate adaptiveness to truly novel safety topics
> *"In the robustness experiment on unseen safety categories, the “Criminal Planning / Confessions” topic can overlap with categories already covered by Llama-Guard (violent crime) and ShieldGemma (dangerous content). To better demonstrate adaptiveness to truly novel topics, I suggest (i) choosing a domain unrelated to safety—e.g., a keyword-mentioning or mathematics discussion task—and (ii) adding an LLM-as-judge baseline (such as GPT-4o) to evaluate safeguard reliability when judgment is delegated to an external model."*
>
> Thanks for raising this point. The initial adaptiveness evaluation dataset WildGuardTest may overlap with the training data of some baseline guard models, potentially favoring them. To mitigate this, we first evaluated on GSM8K, a math dataset unrelated to safety. As expected, all models, including RSafe and its baselines, correctly identified the content as harmless.
>
> To better assess adaptiveness to novel safety topics, we turned to two medical ethics benchmarks: MedSafetyBench [7] and CARES [8], the latter including both base and adversarial versions. As shown in Table 1, OpenAI Moderation, ShieldGemma-9B, LlamaGuard3-8B, and RSafe all show significant drops in accuracy on these datasets compared to their average performance on general safety benchmarks (from Table 1 in original paper). This indicates that MedSafetyBench and CARES present out-of-distribution safety challenges beyond the training domains of all models.
>
> RSafe outperforms its baselines on both datasets. When guided by specific medical principles at inference (RSafe-adaptive), its performance improves notably. Compared to LLM-as-judge (GPT-4o provided with corresponding ethical guidelines), RSafe-adaptive achieves comparable performance, demonstrating strong adaptiveness and reliability as a safety guardrail model.
>
>
>
> Table 1. Adaptiveness comparison of RSafe and baseline models on novel safety topics.
>
> |            | GSM8K prompt | GSM8K response | Weighted Avg (general)             | MedSafetyBench | CARES (base)   |            | CARES(adv)   |              |
> |--------------|----------------|-------------|-------------|------------------|--------------|--------------|--------------|--------------|
> |                       | Acc   | Acc       | Acc         | Acc        | Acc   | F1           | Acc     | F1           |
> | OpenAI Moderation     | 1            | 1              | 0.626             | 0.006            | 0.241        | 0.069        | 0.220        | 0.018        |
> | ShieldGemma-9B        | 0.998        | 1              | 0.702           | 0.288            | 0.475        | 0.500        | 0.324        | 0.247        |
> | LlamaGuard3-8B        | 1            | 1              | 0.877           | 0.644            | 0.646        | 0.711        | 0.382        | 0.354        |
> | **RSafe**             | 1            | 1              | 0.904             | 0.588            | 0.576        | 0.631        | 0.433        | 0.429        |
> | **RSafe-adaptive**    | 1            | 1              | -                | 0.804            | 0.709        | 0.773        | 0.501        | 0.535        |
> | LLM-as-judge (GPT-4o) | 1            | 1              | -                  | 0.881            | 0.632        | 0.738        | 0.456        | 0.527        |
>
>
> References
>
> [1] Liu, Yue, et al. "Guardreasoner: Towards reasoning-based llm safeguards."  (2025).
>
> [2] Rafailov, Rafael, et al. "Direct preference optimization: Your language model is secretly a reward model." (NIPS 2023).
>
> [3] Guo, Daya, et al. "Deepseek-r1: Incentivizing reasoning capability in llms via reinforcement learning." (2025).
>
> [4] Xu, Shusheng, et al. "Is dpo superior to ppo for llm alignment? a comprehensive study." (ICML 2024).
>
> [5] Wu, Junkang, et al. "Towards robust alignment of language models: Distributionally robustifying direct preference optimization." (ICLR 2025)
>
> [6] Wang, Zhichao, et al. "A comprehensive survey of llm alignment techniques: Rlhf, rlaif, ppo, dpo and more." (2024).
>
> [7] Han, Tessa, et al. "Medsafetybench: Evaluating and improving the medical safety of large language models." (NIPS 2024)
>
> [8] Chen, Sijia, et al. "CARES: Comprehensive Evaluation of Safety and Adversarial Robustness in Medical LLMs." (2025).

---

> > ### Comment · Reviewer_Fe4b · 2025-08-06
> > **Response after rebuttal**
> >
> > Thank you for you response. I will keep my score.

---

> > > ### Author Response · Authors · 2025-08-07
> > >
> > > Thanks for your time and your acknowledgement.

---

### Official Review · Reviewer_zR4h · 2025-07-07

**Clarity:** 3
**Significance:** 3
**Originality:** 2
**Rating:** 5
**Confidence:** 4

**Summary:**

One natural approach to model safety is training/constructing an unsafe input/output filter. This paper proposes training the input/output filter using RL instead of the standard fine-tuning approach utilized by prior works. Furthermore, they enable adaptive classification by allowing users to specify content policies in natural text.

**Questions:**

Asked in weaknesses

**Ethical Concerns:**

["NO or VERY MINOR ethics concerns only"]

**Limitations:**

Yes

**Quality:**

3

**Strengths And Weaknesses:**

Strengths
1. The paper framing/motivation is incredibly clear. The writing is also easy to follow.
2. The results/baselines are clearly described, as well as showing string gains in the WildGuardTest setting.

Weaknesses
1. The method lacks some novelty. For example, training an input/output filter and supporting adaptive policies has been introduced in prior works such as LlamaGuard. The main contribution seems to be limited to utilizing RL
2. For the WildGuardTest evaluation, don't the guard models also admit adaptive variants by specifying policies? Do these methods match performance?
3. [Minor] Is it possible to report averaged numbers as well for Table 1 for easier comparison?

---

> ### Author Rebuttal · Authors · 2025-07-30
>
> > ### Comment 1: Adaptive safety policy support has been explored in prior work
>  *"The method lacks some novelty. For example, training an input/output filter and supporting adaptive policies has been introduced in prior works such as LlamaGuard. The main contribution seems to be limited to utilizing RL."*
>
>
> Thanks for bringing this up. As Table 1 shows, some widely adopted guard models like ShieldGemma [1] support user-defined safety policies at inference time, enabling adaptation beyond their default coverage. While RSafe also supports user-specified safety policies during inference, it fundamentally differs from traditional guardrail models. Conventional models rely on supervised fine-tuning to fit safety judgments over in-distribution data. In contrast, RSafe learns to reason about safety during training via safety thinking—instead of merely fitting labels, it internalizes how to make context-aware safety decisions based on provided policies. This enables better adaptivity and interpretability.
>
> To compare the adaptivity of RSafe with traditional counterparts, we employ MedSafetyBench [2], a benchmark constructed around nine medical ethics principles that lie outside the default safety policies seen during training for both ShieldGemma and RSafe. The benchmark comprises 1,800 harmful instructions that explicitly violate these principles. As shown in Table 4, both RSafe and ShieldGemma exhibit substantially lower accuracy on this benchmark compared to their in-distribution performance, confirming that MedSafetyBench constitutes a novel and out-of-distribution safety domain for both models.
>
> When we provided the 9 medical principles at inference time (adaptive version), both models improved significantly. However, RSafe-adaptive continued outperforming ShieldGemma-adaptive, indicating our reasoning-based training paradigm enables stronger adaptiveness to novel safety policies than traditional data-fitting methods. The reasoning trace also provide interpretability for safety decisions.
>
> Table 1. Comparison of RSafe with prevailing guardrail models.
>
> | Guard model     | Accept user specified safety policy    | Accept new safety policy     | Explainability |
> |-------------|------------|-------------|-------------:|
> | OpenAI Moderation        |    ❌    | ❌        |  ❌     |
> | ShieldGemma-9B        |   ✅      |  ✅      | ❌   |
> | LlamaGuard3-8B        |    ❌    |    ❌      |  ❌ |
> | WildGuard-7B         |     ❌    |     ❌     |   ❌ |
> | GuardReasoner-8B       |     ❌    |    ❌     | ✅ |
> | RSafe       |    ✅  |   ✅    | ✅ |
>
>
> Table 2. RSafe vs. ShieldGemma on adaptative capability to novel safety policies.
>
> | Guard model     | MedSafetyBench (Acc ↑)   |
> |-------------|-------------:|
> | ShieldGemma-9B        |   0.288      |
> | ShieldGemma-9B-adaptive       |   0.708   |
> | **RSafe**       |  0.588    |
> | **RSafe-adaptive**       |  **0.804**    |
>
> >### Comment 2: Missing adaptive baseline comparison in robustness evaluation
>  *"For the WildGuardTest evaluation, don't the guard models also admit adaptive variants by specifying policies? Do these methods match performance?“*
>
> Thanks for bringing this up. Since ShieldGemma is the prevailing guardrail model that also supports injecting user-specific safety policies during inference, we included an adaptive version of ShieldGemma in the WildGuardTest benchmark evaluation. We observed that incorporating custom policies led to significant performance improvement over the default policy. However, its performance remained consistently lower than RSafe's. These results further highlight the superior generalization ability of RSafe's reasoning-based approach when handling out-of-distribution safety scenarios, including adversarial jailbreak scenarios.
>
> Table 3. RSafe vs. ShieldGemma for robustness evaluation on WildGuardTest.
>
> | Guard Model             | Overall     |           | Vanilla     |           | Adversarial |           |
> |---------------------|-------------|-----------|-------------|-----------|-------------|-----------|
> |                         | Acc         | F1        | Acc         | F1        | Acc         | F1        |
> | ShieldGemma-9B          | 0.694       | 0.482     | 0.731       | 0.595     | 0.651       | 0.314     |
> | ShieldGemma-9B-adaptive | 0.806       | 0.783     | 0.846       | 0.839     | 0.706       | 0.711     |
> | **RSafe**               | 0.828       | 0.772     | 0.871       | 0.845     | 0.779       | 0.668     |
> | **RSafe-adaptive**      | **0.845**   | **0.823** | **0.911**   | **0.899** | **0.791**   | **0.717** |
>
> >### Comment 3: Report average performance for better comparison
> *"Is it possible to report averaged numbers as well for Table 1 for easier comparison?"*"
>
> Thank you for pointing this out. We have added the weighted average scores in Table 4 and will include them in the revised paper. As shown in Table 4, RSafe achieves competitive performance on general safety benchmarks compared to leading guardrail models.
>
> Table 4. Weighted average accuracy and F1-score across safety benchmarks, provided as a complement to Table 1 (which presents performance comparisons on prompt and response harmfulness detection tasks) in the original paper.
>
> | Model                  | Weighted Acc | Weighted Unsafe F1 |
> |------------------------|--------------|---------------------|
> | **Closed-source APIs** |              |                     |
> | OpenAI Moderation      | 0.626       | 0.214              |
> | GPT-4o                 | 0.644       | 0.460              |
> | **Open-source models** |              |                     |
> | ShieldGemma-9B         | 0.702       | 0.522              |
> | LlamaGuard3-8B         | 0.877       | 0.833              |
> | WildGuard-7B           | 0.893       | 0.884              |
> | **Open-source (+ reasoning)** |       |                     |
> | GuardReasoner-8B       | 0.899       | 0.889              |
> | **Our model**          |              |                     |
> | RSafe w/o rl           | 0.830       | 0.807              |
> | RSafe w/o reason       | 0.846       | 0.813              |
> | **RSafe**              | **0.904**   | **0.889**          |
>
>
> References
>
> [1] Zeng, Wenjun, et al. "Shieldgemma: Generative ai content moderation based on gemma." (2024).
>
> [2] Han, Tessa, et al. "Medsafetybench: Evaluating and improving the medical safety of large language models." (NIPS 2024)

---

> > ### Comment · Reviewer_zR4h · 2025-08-04
> >
> > Thank you for the detailed response and clarifications. I maintain my (positive) rating of 5. It is not lower because there is a method/artifact with comparisons to fair baselines, it is not higher because of the limited novelty and minimal gains.

---

> > > ### Author Response · Authors · 2025-08-05
> > >
> > > Thank you for your comment and your recognition of our work. We present an adaptive safeguard that enables policy-aware, step-by-step safety reasoning, leveraging rule-based reinforcement learning without requiring additional annotated data. Our method is both conceptually promising and empirically effective. The evaluation shows strong performance on in-distribution safety benchmarks and significant improvements over baselines on out-of-distribution novel safety categories.

---

### Note · Authors · 2025-08-14

We sincerely thank all reviewers for their feedback. Overall, the reviews recognize RSafe as an effective safeguard that:
1. Achieves strong performance on safety benchmarks；
2. Adapts seamlessly to user-defined safety policies;
3. Remains robust against out-of-distribution jailbreak scenarios;
4. Provides clear interpretability for safety decisions.

These strengths are supported by comprehensive experimental validation. Reviewers zR4h and Fe4b further praised our work for its strong motivation, clear framing, and easy-to-following  presentation. During the rebuttal stage, we conducted additional experiments and analyses to address reviewers' concerns. The most notable additions include:

1. **Adaptiveness to truly novel safety topics (Addressing Fe4b comment 2 and A8Ug comment 3).**
We evaluated RSafe on two medical ethics benchmarks, demonstrating strong adaptiveness and reliability as a safety guardrail model when addressing novel safety domains.
2. **Robustness on edge cases / adaptiveness on highly specific safety rules (Addressing A8Ug comment 1).**
We tested RSafe on challenging edge-case prompts from red-teaming datasets and evaluated its performance with highly specific, user-defined safety policies, demonstrating RSafe's comparable robustness and adaptiveness to GPT-4o.

>### Response to Reviewer m19C's Main Concerns

We provided detailed responses to the three primary concerns raised by reviewer m19C in the final rebuttal round, and we re-emphasize it in a summary as follows:

1. **Binary (0/1) Outcome Rewards in GRPO**.
The absence of intermediate-step supervision is an advantage rather than a limitation. This approach mitigates label noise, reduces annotation burden, and enhances overall robustness. The superiority of our experimental results, combined with consistency between reasoning paths and final predictions verified through manual inspection, further validates the robustness of our training framework.

2. **Effectiveness Compared to GuardReasoner-8B**.
We conducted controlled comparisons using matched backbones and baselines, which demonstrated RSafe's superior performance under equivalent experimental conditions.

3. **Reliance on User-Specified Categories**.
RSafe offers an adaptive mode (RSafe-adaptive) for user-defined safety policies while also performing strongly in its default general-policy setting (as demonstrated in the original paper Table 1, and our first-round rebuttal response to m19C, comment 3).

---

### Decision · Program_Chairs · 2025-09-17

**Decision:**

Accept (poster)

**Comment:**

The paper proposes RSafe, a reasoning-based safeguard that integrates policy-guided reasoning with GRPO alignment for adaptable LLM safety moderation. Three reviewers found the work technically solid, clearly presented, and empirically validated, with strong results on out-of-distribution and adversarial settings; two explicitly recommended acceptance, and one raised their score after rebuttal. The authors strengthened the case with new experiments on medical ethics and red-teaming benchmarks, comparisons against GPT-4o, and alternative backbones, which addressed most concerns about robustness and adaptiveness. The remaining critical review (m19C) judged the contribution incremental and less effective than GuardReasoner, but this view was not shared by the majority. Given the overall positive reception and substantial empirical value despite ongoing debate about novelty, I recommend acceptance.